# Beyond single support calls: Efficacy of follow-up callbacks for dementia caregivers using the Alzheimer's association helpline

Nancy A. Hodgson[1]*, Sonia Talwar[1], Subhash Aryal[1], Kerry Finegan[2], Sam Fazio[2]

1 University of Pennsylvania, School of Nursing, Philadelphia, Pennsylvania, United States of America,
2 Alzheimer's Association, Chicago, Illinois, United States of America

* hodgsonn@nursing.upenn.edu

## Abstract

### Background

Family care partners of persons living with dementia (PLWD) have limited support and resources and experience high levels of stress and burden. Prior research has demonstrated that one call to the Alzheimer's Association's Helpline benefitted 80% of callers following a single call. However, some care partners need more support than a single call. The aim of this study was to evaluate the efficacy of offering up to two additional Helpline care consultations after an initial call to the Helpline.

### Methods

2503 study participants were Helpline callers who received one, two, or three care consultations at the callers' request. The PROMIS measure of self-efficacy in managing emotions was assessed among participants over time (baseline to two weeks after each care consultation), and differences were assessed between caller groups.

### Results

No statistical differences were noted in demographics between those who received a single call versus two or three calls. Baseline self-efficacy scores were significantly lower in those who requested more than one call (p < 0.01). Self-efficacy scores improved over time in all three groups of callers, and the improvement was significant in those who received one and two care consultations (p < 0.01).

### Conclusions

Further research is indicated to understand the specific support needs of care partners that can benefit from more than one call to a telesupport helpline.

**Data availability statement:** All relevant data are within the manuscript and its Supporting Information files. The data file ID numbers which were NOT identifiers have been updated to small numbers in numeric order just to be more clear given your request. Any personalized identifiers have been removed. This data set is fully anonymized.

**Funding:** NH, ST, and SA performed research and analysis that was funded by the Alzheimer's Association (https://www.alz.org/). KF and SF are employees of the Alzheimer's Association and are co-authors on this manuscript. KF and SF worked with NH to design and execute this study. KF assisted with data collection.

**Competing interests:** The authors have declared that no competing interests exist. KF and SF are employed by the Alzheimer's Association.

**Abbreviations:** PLWD, person(s) living with dementia; SD, standard deviation.

## Background

Persons living with dementia (PLWD) experience progressive cognitive decline affecting memory, executive function, and communication, often accompanied by behavioral and psychological symptoms such as agitation, wandering, paranoia, and sundowning. These manifestations create extraordinary demands on family care partners, who frequently report severe emotional strain from witnessing personality changes, physical exhaustion from providing round-the-clock supervision, social isolation as care responsibilities intensify, financial hardship from care costs or reduced employment, and significant stress from complex care decisions balancing safety against autonomy [1] Family care partners of PLWD often have limited support and resources and experience high levels of stress and burden. Resources such as telephone support (telesupport) programs provide easy-to-access and anonymous services to support the mental health and well-being of community populations [2–4]. Telephone support lines for family care partners can offer a combination of counseling, information, and referral to services, and has been shown to be effective in improving burden, depression, social support, and increasing knowledge and use of community services [3–5].

Increasingly, psychosocial telehealth and specifically telephone support programs are being evaluated to assess their ability to address the emotional needs of care partners for persons living with dementia [6–8]. The Alzheimer's Association has a free 24/7 national telephone Helpline staffed with trained care consultants. Prior research has found that 80% of callers benefitted from a single helpline care consultation, but that some care partners needed additional support [9]. A 2017 study addressed the challenges of delivering advice via helplines, suggesting that the dose of these calls may need to be evaluated [10]. While single calls can have a positive impact, the structure and content of these calls may need to be carefully considered to ensure their effectiveness. Unfortunately, the prior literature is scant in details regarding dose and duration of calls required to achieve positive outcomes. Evaluations are therefore essential to ensure that helplines can address the needs of multiple types of callers and leverage these learnings to maximize the effectiveness of their services.

The purpose of this study was to evaluate the efficacy of offering up to two additional care consultations after an initial call to the Helpline by addressing two key evaluation questions: First, are there baseline differences in callers who request 1, 2, or 3 care consultations with the Helpline? Second, does each group of callers benefit from the additional care consultations received?.

## Methods

### Research design

This pre-post design study compared study participants who received one, two, or three care consultation calls from the Alzheimer's Association Helpline. Study participants were care partners of PLWD who called the Helpline. Participants were excluded if they: (1) presented with a crisis situation, (2) were frequent repeat callers,

(3) had multiple people on the line, or (4) did not speak English. Crisis situations were defined as circumstances where someone was at risk of harm to themselves or others, including immediate safety concerns, active wandering incidents, or potential medical emergencies requiring immediate attention. Frequent callers were those with an Individual Treatment Plan on file who met one or more of the following criteria: excessive calling about similar non-crisis issues; excessive calling about non-dementia related issues; demonstrated inability, difficulty, or refusal to follow through on provided resources; requiring unusual containment or redirection efforts; using aggressive or inappropriate language toward staff; or disclosing existing, regular relationships with medical or mental health professionals.

All non-crisis callers to the Alzheimer's Association Helpline between July 2021 and July 2022 were asked to participate in this study and the Helpline agent obtained verbal consent during the call and documented the consent response in their internal data collection system (Personify). Callers were then asked for demographics at baseline and the Helpline agent administered the baseline survey questions. Then, the Helpline agent transferred the caller to a trained Alzheimer's Association care consultant to conduct or schedule a care consultation. All care consultations conducted lasted up to one hour in duration and resulted in the creation of an action plan. Together, care consultants and callers created an action plan consisting of reasonable goals and next steps for the care partner to engage in to help address any issues and concerns discussed during the care consultation. Two weeks (14 days) after the care consultation, a research specialist from the Alzheimer's Association conducted a follow-up survey. These questions asked if the caller was able to complete any of the action plan steps, if they accessed any resources discussed, and how helpful they found the action plan. If participants exhibited no progress towards action steps, mentioned barriers to completing action steps, and/or talked about new concerns and/or needs, they were asked if they wanted an additional care consultation. If the participant met any or multiple conditions, they were transferred to an available care consultant for a second care consultation or to schedule a second care consultation. Two weeks (14 days) after the second care consultation, the research specialist conducted a second follow-up survey, which was identical to the first follow-up survey. Again, participants were asked if they would like an additional care consultation if they showed no progress towards action steps, barriers to completing action steps, and/or new concerns and/or needs. If the participant agreed to a third care consultation, they were either transferred to an available care consultant or scheduled a care consultation. The third and final follow-up survey was administered by the research specialist two weeks (14 days) after the third care consultation. This follow-up survey was the same as the first two follow-up surveys. This concluded the care partner's participation in the study. Care partners who completed all surveys received a $20 gift card. All study and consent procedures were approved by the University of Pennsylvania IRB (Protocol # 849227).

### Measures

The baseline survey items consisted of basic demographic questions (care partner and PLWD age, race, gender, and years of education) collected at baseline only. Additional demographic information included PLWD living situation and diagnosis, and care partner role. The baseline and follow-up surveys included the primary outcome, the PROMIS Self-Efficacy in Managing Emotions Short Form 8A (Cronbach's alpha 0.90–0.95), which assessed caregiver confidence in their ability to manage fatigue, pain, emotional distress, and other symptoms using self-management techniques [11,12]. The eight PROMIS items were scored on a Likert scale ranging from 1 (not at all confident) to 5 (very confident). A higher summed PROMIS score indicated higher self-efficacy in managing emotions. The PROMIS Self Efficacy instrument is well-validated, with all versions demonstrating high internal consistency and reliability [11] (Gruber-Baldini et al., 2017). It has shown effectiveness in evaluating individuals' ability to cope with stress across diverse populations and disease trajectories. The follow-up surveys additionally asked whether callers had implemented the "action plan", whether they had accessed any support services recommended in the action plan, how helpful they found the action plan, if they understood the action plan, and to discuss any barriers to completing the action plan steps.

## Statistical analysis

Data is presented as mean (standard deviation, SD) for continuous variables or frequency (proportion) for categorical variables. We determined the normal distribution assumption for continuous variables graphically using boxplots and the Shapiro-Wilk test. We did not perform *a priori* power calculation.

First, we compared the three groups of callers on baseline characters. To evaluate the differences in care partner and PLWD age between participants who received 1 follow-up call, 2 follow-up calls, or 3 follow-up calls we performed one-way analysis of variance (ANOVA) followed by Tukey test for each pairwise comparison. Similarly, for comparing categorical characteristics between groups (1 follow-up call, 2 follow-up calls, or 3 follow-up calls) we used either chi-square test or Fisher's exact test when the expected cell-frequency was less than 5.

Next to evaluate within person change over time, we performed paired *t*-test for normally distributed data and Signed-Rank test when the data did not follow normal distribution. To protect from inflated type I error due to multiple comparison, we used Bonferroni adjusted type I error rate = 0.01. To confirm the robustness of our findings, we performed sensitivity analysis. For items with missing data, we imputed values to obtain complete data for all PROMIS items before calculating the PROMIS T-scores and repeated our comparisons using the imputed data. All analysis and imputation were conducted using SAS 9.4.

## Results

### Study cohort and characteristics

As shown in Table 1, 2503 adult participants participated in the survey at baseline and each subsequent wave of the survey. 97.68% of the callers were family care partners, 79.92% were female, and 73% were white. The average age of participants was 60.31 years (± 12.59 years), 85.06% possessed a bachelor's degree or higher, and 64.42% lived with the PLWD. The cohort of participants receiving one Helpline care consultation and follow-up consisted of 1375 participants. Similarly, 83 participants received two Helpline care consultations and 13 participants received three care consultations after baseline survey. There was no statistical difference in demographics between callers who received one call versus callers who received more than one call.

### PROMIS Total (T) Score Analysis

Participants who received 1 follow-up call had a mean (SD) baseline PROMIS total score of 47.1 (6.4). The mean (SD) baseline PROMIS total score for the 83 participants who received 2 follow-up calls was 44.5 (6.4). Finally, the 13 participants who received 3 follow-up calls had mean (SD) baseline PROMIS total score of 39.4 (5.1). One-way ANOVA revealed a significant difference in the PROMIS total score at baseline between the 1 follow-up, 2 follow-up, and 3 follow-up calls groups (p < 0.0001). Pairwise comparison using Tukey test revealed significant difference between 1 follow-up vs 2 follow-up and 1 follow-up vs 3 follow-up groups but no difference between 2 follow-up vs 3 follow-up groups. The mean differences and the corresponding 95% CI for mean differences are presented in Table 2.

Further, to evaluate change over time, we performed paired data analysis between multiple waves of surveys. The PROMIS total score increased at first follow-up compared to baseline by 0.47 points, and the PROMIS total score was higher by 1.67 at the second follow-up compared to baseline. The changes were statistically significant (p < 0.01). Among the 13 participants who participated in the baseline survey and received 3 subsequent follow-up calls, there was no significant change in their PROMIS T-score compared to baseline (p > 0.01). The 87 people who received a second follow-up call after baseline improved their PROMIS total score by 1.29 points compared to the score obtained after receiving the first follow-up. The increase was statistically significant (p < 0.01). In Table 3, we present the mean (SD) difference and the corresponding p-values for pairwise comparison for the overall PROMIS total score and the individual item scores. Five key items from the PROMIS Self-Efficacy measure showed significant improvements. For participants who received one

**Table 1. Comparison of demographic characteristics.**

| Variable | Overall $n=2503$ | Baseline + 1 Other Call $n=1375$ | Baseline + 2 Other Calls $n=83$ | Baseline + 3 Other Calls $n=13$ | p-value |
|---|---|---|---|---|---|
| Care Partner Role | | | | | 0.1821 |
| Family | 2445 (97.68%) | 1347 (97.96%) | 80 (96.39%) | 12 (92.31%) | |
| Friend | 51 (2.04%) | 24 (1.75%) | 3 (3.61%) | 1 (7.69%) | |
| Professional | 7 (0.28%) | 4 (0.29%) | 0 (0%) | 0 | |
| Care Partner Education | | | | | 0.3943 |
| High School or Less | 367 (14.66%) | 179 (13.02%) | 16 (19.28%) | 2 (15.38%) | |
| Bachelor's or higher | 2129 (85.06% | 1191 (86.62%) | 67 (80.72%) | 11 (84.62%) | |
| Declined | 7 (0.28%) | 5 (0.36%) | 0 | 0 | |
| Care Partner Ethnicity | | | | | 0.2057 |
| Non-White | 675 (26.97%) | 349 (25.38%) | 27 (32.53%) | 5 (38.46%) | |
| White | 1828 (73.03%) | 1026 (74.62%) | 56 (67.47%) | 8 (61.54%) | |
| Care Partner Gender | | | | | 0.1310 |
| Female | 1993 (79.62%) | 1105 (80.36%) | 62 (74.70%) | 8 (61.54%) | |
| Male | 501 (20.02%) | 266 (19.35%) | 20 (24.10%) | 5 (38.46%) | |
| Missing | 9 (0.36%) | 4 (0.29%) | 1 (1.20%) | 0 | |
| Care Partner Age | $n=2484$ | $n=1362$ | $n=82$ | $n=13$ | 0.1759 |
| Mean (SD) | 60.5 (12.6) | 61.2 (12.5) | 63.9 (12.3) | 61.9 (13.9) | |
| PLWD Living Situation | | | | | 0.3418 |
| Not with Family | 891 (35.60%) | 497 (36.15%) | 25 (30.12%) | 3 (23.08%) | |
| With Family | 1612 (64.40%) | 878 (63.85%) | 58 (69.88%) | 10 (76.92%) | |
| PLWD Education | | | | | 0.4798 |
| High School or Less | 1153 (46.06%) | 615 (44.73%) | 36 (43.37%) | 6 (46.15%) | |
| Bachelor's or higher | 1306 (52.18%) | 737 (53.60%) | 46 (55.42%) | 6 (46.15%) | |
| Declined | 44 (1.76%) | 23 (1.67%) | 1 (1.20%) | 1 (7.69%) | |
| PLWD Ethnicity | | | | | 0.6872 |
| Non-White | 676 (27.01%) | 347 (25.24%) | 24 (28.92%) | 4 (30.77%) | |
| White | 1827 (72.99%) | 1028 (74.76%) | 59 (71.08%) | 9 (69.23%) | |
| PLWD Gender | | | | | 0.1175 |
| Female | 1540 (61.53%) | 836 (60.80%) | 43 (51.81%) | 10 (76.92%) | |
| Male | 944 (37.71%) | 532 (38.69%) | 40 (48.19%) | 3 (23.08%) | |
| Missing | 19 (0.76%) | 7 (0.51%) | 0 | 0 | |
| PLWD Diagnosis | | | | | 0.013 |
| Alzheimer | 914 (36.52%) | 536 (38.98%) | 27 (32.53%) | 1 (7.69%) | |
| Dementia Suspected | 485 (19.38%) | 241 (17.53%) | 20 (24.10%) | 7 (53.85%) | |
| Dementia Unspecified | 763 (30.48%) | 398 (28.95%) | 28 (33.73%) | 4 (30.77%) | |
| Other Diagnosis | 341 (13.62%) | 200 (14.54%) | 8 (9.64%) | 1 (7.69%) | |
| PLWD Age | $n=2463$ | $n=1356$ | $n=83$ | $n=13$ | 0.9239 |
| Mean (SD) | 79.14 (9.0) | 79.2 (9.1) | 78.8 (8.6) | 78.9 (9.6) | |

Note: *p*-value from Chi_Square Test or Fisher's Exact Test for Categorical Variable; One-way ANOVA for Continuous.

 

**Table 2. PROMIS T-score at baseline.**

| Variable | Received 1 Call after Baseline n=1375 | Received 2 Calls after Baseline n=83 | Received 3 Call after Baseline n=13 | Mean Difference 95% CI |
|---|---|---|---|---|
| PROMIS T Score | n=1257 | n=79 | n=11 | 1 vs 2, 2.5 (0.6, 4.4)*** |
| Mean (SD) | 47.1 (6.4) | 44.5 (6.4) | 39.4 (5.1) | 1 vs 3, 7.6 (2.5, 12.7)*** 2 vs 3, 5.1 (−0.2,10.4) |

**Table 3. Paired Comparison between groups of callers.**

| Variable | 1st Follow-Baseline | 2nd Follow-Baseline | 3rd Follow-Baseline | 2nd Follow-1st Follow | 3rd Follow-2nd Follow |
|---|---|---|---|---|---|
| PROMIS Total Score | N=1266 | N=96 | N=11 | N=87 | N=13 |
| | Mean (SD) = 0.47 (4.7) | Mean (SD) = 1.67 (5.6) | Mean (SD) = 0.17 (3.5) | Mean (SD) = 1.29 (4.3) | Mean (SD) = −1.05(5.4) |
| | P=0.0003* | P=0.0039* | P=0.8203 | P=0.0098* | P=0.7334 |
| Treat | N=1274 | N=96 | N=11 | N=87 | N=13 |
| | Mean (SD) = −0.06 (1.0) | Mean (SD) = −0.07 (1.1) | Mean (SD) = −0.81 (0.9) | Mean (SD) = 0.16 (1.0) | Mean (SD) = −0.30 (1.1) |
| | P=0.0306 | P=0.6292 | P=0.0313 | P=0.1916 | P=0.2813 |
| Feels | N=1467 | N=96 | N=13 | N=96 | N=13 |
| | Mean (SD) = 0.07 (0.9) | Mean (SD) = 0.18 (1.2) | Mean (SD) = 0.46(0.8) | Mean (SD) = 0.01(0.8) | Mean (SD) = −0.30(1.3) |
| | P=0.0022* | P=0.1435 | P=0.1563 | P=0.8871 | P=0.6563 |
| Stress | N=1468 | N=96 | N=13 | N=96 | N=13 |
| | Mean (SD) = 0.08 (0.9) | Mean (SD) = 0.29 (1.1) | Mean (SD) = 0.30 (0.9) | Mean (SD) = 0.18 (1.0) | Mean (SD) = 0.07 (1.1) |
| | P=0.0011* | P=0.0094* | P=0.3984 | P=0.0666 | P=0.9999 |
| Upset | N=1467 | N=96 | N=13 | N=96 | N=13 |
| | Mean (SD) = 0.06 (0.9) | Mean (SD) = 0.27 (1.2) | Mean (SD) = 0.15 (1.1) | Mean (SD) = 0.16 (0.7) | Mean (SD) = −0.76 (1.1) |
| | P=0.0081* | P=0.0291 | P=0.5625 | P=0.0437 | P=0.0313 |
| Avoid | N=1465 | N=96 | N=13 | N=96 | N=13 |
| | Mean (SD) = 0.07(0.9) | Mean (SD) = 0.22(1.0) | Mean (SD) = 0.07(0.9) | Mean (SD) = 0.07(1.0) | Mean (SD) = 0.46(0.8) |
| | P=0.0033* | P=0.0552 | P=0.9999 | P=0.5887 | P=0.1563 |
| Keep | N=1466 | N=96 | N=13 | N=96 | N=13 |
| | Mean (SD) = 0.15(1.0) | Mean (SD) = 0.33(1.1) | Mean (SD) = 0.30(0.7) | Mean (SD) = 0.16(1.0) | Mean (SD) = −0.07 (1.3) |
| | P=0.0001* | P=0.0113 | P=0.3125 | P=0.1017 | P=0.9375 |
| Bounce Back | N=1464 | N=96 | N=13 | N=96 | N=13 |
| | Mean (SD) = 0.02 (0.9) | Mean (SD) = 0.13 (1.0) | Mean (SD) = −0.46(0.7) | Mean (SD) = 0.17 (0.9) | Mean (SD) = −0.30(1.2) |
| | P=0.3395 | P=0.2342 | P=0.1094 | P=0.0438 | P=0.4688 |
| Relax | N=1466 | N=96 | N=13 | N=96 | N=13 |
| | Mean (SD) = 0.04 (1.0) | Mean (SD) = 0.31 (1.3) | Mean (SD) = 0.46(1.12) | Mean (SD) = 0.25 (1.0) | Mean (SD) = 0.07(1.0) |
| | P=0.1312 | P=0.0371 | P=0.2500 | P=0.0236 | P=0.9844 |

Note: P-value reported for 1st Follow-Baseline is from Student's t-test and the rest of the p-values are from Signed Rank test. To adjust for multiple comparison using Bonferroni approach, only p-values<0.01 are considered statistically significant.

care consultation, four items showed significant improvement (p<0.01): "I can handle negative feelings," "I can find ways to manage stress," "I can avoid feeling discouraged [by] stress," and "I can keep emotional distress from interfering with things." For participants who received two care consultations, only the item "I can find ways to manage stress" showed significant improvement from baseline (p<0.01).

We conducted a sensitivity analysis using complete data by imputing missing individual items before calculating the overall PROMIS total score. We repeated the analysis described above for PROMIS total scores. The findings from our sensitivity analysis were similar to the observed data analysis and we only present results from the observed data analysis. We did not conduct a sensitivity analysis for individual PROMIS items.

## Discussion

The objectives of this study were to compare baseline differences in callers who requested one, two, or three care consultations with the Alzheimer's Association Helpline, and to examine whether callers benefit from the care consultations received. Our first finding indicates that those who later requested additional calls had the lowest levels of self-efficacy in managing emotions at baseline, suggesting an elevated level of need. It is important to note that the PROMIS Self-Efficacy in Managing Emotions total scores have a normative mean of 50 with SD of 10 in the United States general population [11]. Compared to that normative data, caregivers who called the Helpline had below-average emotional self-efficacy, especially when requesting multiple consultations (one: 47.1, two: 44.5, three: 39.4).

This pattern suggests a potential "dose-response" relationship between caregiver distress and help-seeking behavior. Caregivers experiencing greater emotional challenges may recognize their need for additional support and proactively seek it through multiple consultations. Alternatively, helpline staff may identify these individuals as requiring more intensive follow-up based on their initial presentation. This finding aligns with the broader literature on caregiver burden in dementia, which consistently demonstrates that emotional exhaustion and diminished self-efficacy are primary drivers of formal support utilization [13–14].

Our second finding suggests that callers benefitted on average as demonstrated by improving self-efficacy scores at all follow-ups. The findings were statistically significant for those in the one call and two call groups. PROMIS items around managing negative feelings and stress improved significantly after one and two care consultations. Small sample size limited the ability to test for significance in the three-call group. These findings about the effectiveness of the Helpline's care consultations are aligned with the findings of recent evaluations of the Helpline. In a 2023 study, telephone health visits were preferred by callers compared to video health visits, and telephone visits were seen to be shorter and tended to cover more material [15]. Telephone counseling was found to have the same level of high interpersonal quality as in-person counseling and was elected more often than in-person counseling in another 2023 study [16]}. Tsui et al., [17] reported that phone-based support calls for home health aides of PLWD resulted in improved communication and empathy.

The improvement in self-efficacy scores across consultations suggests that telephone-based interventions can effectively address caregiver emotional needs even without face-to-face interaction. This finding is particularly relevant given the accessibility barriers many dementia caregivers face, including geographic isolation, transportation limitations, and inability to leave care recipients unattended. The specific improvements in items related to managing negative feelings and stress highlight the importance of emotional regulation skills in the caregiving context. These targeted benefits may result from the helpline staff's training in validating caregiver experiences while providing concrete coping strategies tailored to dementia-specific challenges.

It is noteworthy that certain PROMIS items showed more significant improvement than others, suggesting that telephone consultations may be particularly effective for specific aspects of emotional self-efficacy. Future research might explore which components of emotional regulation are most responsive to telephone intervention and which may require more intensive or different modes of support. Additionally, the differential response patterns across the one-call, two-call, and three-call groups warrant further investigation to determine optimal "dosing" of telephone support for caregivers with varying needs.

### Strengths and limitations

This study has several limitations. First, there are concerns regarding the representativeness of the sample. Participants were those who consented to research, and the anonymous nature of the Helpline prohibits demographic comparisons between

study participants and other callers to the Helpline. However, the demographic characteristics of the sample are comparable to the general profile of people accessing the Helpline. Regardless, Helpline callers may not be representative of the larger population of dementia care partners as the sample was largely white and well-educated. An additional limitation is the loss-to-follow-up when reaching out to callers to complete surveys after they had received a care consultation. Those engaging with subsequent calls may be a self-selected group who find benefit from the consultation calls. Qualitative feedback from participants, while unavailable to us, would better illustrate caller concerns and perceived benefits of the care consultations.

The self-selection bias is particularly relevant when interpreting the outcomes of multiple consultations, as those who experienced little benefit from initial calls would be unlikely to pursue additional support. Furthermore, our study design cannot account for confounding variables that might influence both help-seeking behavior and emotional self-efficacy, such as disease progression in the care recipient, changes in the caregiving situation, or concurrent utilization of other support services. The reliance on self-reported measures also introduces potential response bias, particularly among caregivers who may feel pressure to report improvement after receiving support.

The timing of the follow-up assessments represents another limitation, as the variable intervals between consultations and measurements make it difficult to establish clear temporal relationships between interventions and outcomes. Additionally, while the PROMIS Self-Efficacy measure is well-validated, it may not capture all relevant dimensions of caregiver well-being, such as physical health impacts, relationship quality with the care recipient, or practical caregiving competence.

Despite these limitations, this study also has important strengths. This evaluation fills gaps in the research regarding the number of calls that callers to helplines may need in order to achieve a benefit and adds to our understanding of the benefits of telephone-based supports for dementia care partners. The use of a standardized, validated measure allows for comparison with normative data and other caregiver interventions. The naturalistic design captures real-world utilization patterns and outcomes, enhancing ecological validity and practical applicability of findings. Furthermore, the focus on emotional self-efficacy addresses a critical but often overlooked dimension of caregiver support, as many interventions emphasize knowledge and skills at the expense of emotional coping capacity.

### Implications and future directions

Practice recommendations cannot be determined from the findings of this quasi-experimental study. However, several important implications emerge. First, telephone helplines appear to be a viable intervention for improving emotional self-efficacy among dementia caregivers, particularly those with moderate levels of distress. Second, the pattern of decreasing baseline self-efficacy across groups suggests that helplines might benefit from implementing systematic screening to identify callers with lower emotional self-efficacy who may require more intensive or specialized support. Third, the significant improvements in specific aspects of emotional regulation point to potential mechanisms of action that could be enhanced in future telephone support protocols.

Future evaluations could compare different models of screening callers for level of support or level of need, and use this information to develop care consultation programs aligned with caller needs that provide benefits in the most cost-effective ways. With increasing emphasis on the delivery of best-practice care, building the evidence base in this field may assist care partner helplines to increase their service uptake, reach, and benefits to callers. Additional research directions include examining the long-term sustainability of improvements in emotional self-efficacy, identifying the specific components of telephone consultations that drive these improvements, and exploring how telephone support might be optimally integrated with other service modalities. Investigating potential moderators of treatment response, such as caregiver relationship to the person with dementia, stage of disease, presence of behavioral symptoms, and cultural factors, would help tailor interventions to diverse caregiver populations. Finally, economic analyses comparing the cost-effectiveness of telephone support to other intervention modalities would provide valuable information for healthcare systems and policy-makers seeking to address the growing public health challenge of dementia caregiving.

## Conclusions

Helplines are an integral support to care partners of PLWD. The demand for supports tailored to this group is anticipated to increase as the diversity of care partners expands. Our findings suggest that callers with low levels of self-efficacy may require more than one care consultation. Though our study demonstrated telephone helpline effectiveness for these callers, multiple predictors likely influence intervention outcomes that warrant investigation in future research, including caregiver characteristics (relationship to PLWD, age, education), care recipient factors (dementia type and stage, symptom severity), existing support utilization, geographic accessibility of services, cultural attitudes toward caregiving, and help-seeking patterns. Understanding these variables would enhance our ability to tailor telephone support interventions to the distinct needs of dementia care partners facing these multifaceted challenges. Further research is indicated to understand the support needs of care partners that can benefit from more than one call to a telesupport program.

## Supporting information

**S1 File. IRB Approval Letter.**
(DOCX)

**S2 File. Helpline Data-Manuscript Version-5.**
(XLSX)

## Acknowledgments

We would like to thank Felicia Brown, Terri Krallitsch, and Tim Tully from the Alzheimer's Association for their support on this project.

## Author contributions

**Conceptualization:** Nancy A. Hodgson, Sam Fazio.

**Data curation:** Nancy A. Hodgson, Kerry Finegan.

**Formal analysis:** Nancy A. Hodgson, Subhash Aryal.

**Funding acquisition:** Nancy A. Hodgson.

**Investigation:** Nancy A. Hodgson, Sam Fazio.

**Methodology:** Nancy A. Hodgson, Subhash Aryal, Sam Fazio.

**Project administration:** Nancy A. Hodgson, Sonia Talwar, Kerry Finegan.

**Resources:** Kerry Finegan, Sam Fazio.

**Supervision:** Nancy A. Hodgson, Sonia Talwar, Kerry Finegan, Sam Fazio.

**Validation:** Nancy A. Hodgson, Kerry Finegan, Sam Fazio.

**Visualization:** Nancy A. Hodgson.

**Writing – original draft:** Nancy A. Hodgson, Sonia Talwar, Subhash Aryal.

**Writing – review & editing:** Nancy A. Hodgson, Sonia Talwar, Subhash Aryal, Kerry Finegan, Sam Fazio.

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
