## [Decision Letter · Decision Letter 0]

27 Feb 2025

Dear Dr. Hodgson,

Thank you for submitting your manuscript to PLOS ONE. After careful consideration, we feel that it has merit but does not fully meet PLOS ONE’s publication criteria as it currently stands. Therefore, we invite you to submit a revised version of the manuscript that addresses the points raised during the review process.

We look forward to receiving your revised manuscript.

Kind regards,

Vineet Gupta, MD, FACP, SFHM, CHCQM

Academic Editor

PLOS ONE

Journal Requirements:

2. In the ethics statement in the Methods, you have specified that verbal consent was obtained. Please provide additional details regarding how this consent was documented and witnessed, and state whether this was approved by the IRB

 “NH, ST, and SA performed research and analysis that was funded by the Alzheimer's Association (https://www.alz.org/). KF and SF are employees of the Alzheimer's Association and are co-authors on this manuscript. KF and SF worked with NH to design and execute this study. KF assisted with data collection.”

“The authors have declared that no competing interests exist. KF and SF are employed by the Alzheimer’s Association.”

We note that one or more of the authors are employed by a commercial company: Alzheimer’s Association

6. We note that you have indicated that there are restrictions to data sharing for this study. PLOS only allows data to be available upon request if there are legal or ethical restrictions on sharing data publicly. For more information on unacceptable data access restrictions, please see http://journals.plos.org/plosone/s/data-availability#loc-unacceptable-data-access-restrictions.

Additional Editor Comments:

See above

Reviewers' comments:

Reviewer's Responses to Questions

**Comments to the Author**

1. Is the manuscript technically sound, and do the data support the conclusions?

Reviewer #1: Yes

Reviewer #2: Partly

2. Has the statistical analysis been performed appropriately and rigorously?

Reviewer #1: Yes

Reviewer #2: Yes

3. Have the authors made all data underlying the findings in their manuscript fully available?

Reviewer #1: No

Reviewer #2: No

4. Is the manuscript presented in an intelligible fashion and written in standard English?

Reviewer #1: Yes

Reviewer #2: Yes

Reviewer #1: As the population of those living with dementia increases in size, learning how to effectively support caregivers is worthwhile. In this manuscript, Hodgson et al. explore the value in offering multiple phone consultations to those caregivers of persons living with dementia. The authors collect data on a large sample of caregivers. The main measure of interest is self-efficacy ratings for managing negative feelings (PROMIS T-score): Low scores are associated with accepting more than one consultation, and on average consultations coincide with an improvement in those self-efficacy ratings.

Population trends (more people living with dementia) make this a reasonably timely report, and, I imagine, the report is of adequate interest to the readership of PLoS ONE. I have comments that I hope will improve the manuscript, listed in roughly the same order as the manuscript. The possibility of attrition bias deserves the most attention (see #5 and #6). It seems inevitable for this experimental design coupled with the described recruitment method. There are probably artful future-oriented ways to highlight this for the reader (What would next steps be, given the present findings?)

(1) Exclusions are (or seem) intuitive, but it would be nice to know how many were excluded, and/or to “operationalized/define “crisis situation” and “frequent repeat caller” a bit more. If this is left to the discretion of the Helpline agent, then none of that information may be available, but a sentence/phrase with that explanation could be added to the manuscript.

(2) The first paragraph of the discussion could probably be tightened up a bit: “A score of 40 is 1 standard deviation lower than US general population and score of 60 is 1 SD higher than the US general population.” is redundant with the immediately preceding sentence and could just be removed. The subsequent three sentences, “Callers who requested three calls […]” could be condensed something like this: “Compared to that normative data, out caregivers had below-average emotional self-efficacy, especially when requesting multiple consultations (one: 47.1, two: 44.5, three: 39.4).”

(3) “Our first finding indicates that those who later requested additional calls had the lowest levels of self-efficacy in managing emotions at baseline, indicating the highest level of need.” My sense is that “highest level of need” extends beyond that PROMIS can measure. There is more to “need” than emotional self-efficacy, after all. Something like “[…] suggesting an elevated level of need” might better capture the scope of the present work.

(4) This paragraph is more of a result than a discussion; the authors should consider moving to another section, or abbreviating or eliminating it, since the information it contains is a retelling of the contents of Table 3:

“Five items from the PROMIS Self-Efficacy measure were significant in those who had one or two care consultations. The items “I can handle negative feelings”, “I can find ways to manage stress”, “I can avoid feeling discouraged stress”, and “I can keep emotional distress from interfering with things” were significantly improved (p <0.01) in those who had one care consultation. The item “I can find ways to manage stress” was significantly improved from baseline in those who had two care consultations” (p<0.01).”

(5) “This study has several limitations. First […]” The authors name only one limitation before proceeding to the next paragraph (“Despite these limitations […]”). …did the authors intend for the later paragraph (“Practice recommendations cannot […]”) to be a second limitation? Another one would be the loss-to-follow-up of almost half the sample (2503 with initial consultation, 1375 of those engaging with the call two weeks later). Another would be that those engaging with subsequent calls may be a self-selected group of people who find benefit from the consultation calls.

(6) “Our second finding suggests that callers benefited from all calls […]” are the authors sure they want to commit to *all*? Every single one? What if some of those 2503 didn’t value the call, and that’s why they were lost to follow-up?

Reviewer #2: The authors have taken up a socially important, cognitively original topic that can supplement an important area of ignorance in the field of designing psychoeducational and support activities for families of people with dementia (PLWD). Thus, the research undertaken is part of an important discourse on the needs visible in the healthcare system and ensuring the psychological well-being of caregivers with mental disorders.

The manuscript submitted for evaluation provides interesting data that have application value and can be used to design systemic solutions. In my opinion, however, the text requires the introduction of necessary changes that can improve its quality.

First of all, I propose expanding the theoretical background. Although there are few comparable studies, it would be worthwhile to outline in a more exhaustive way the difficult and problematic situations that result from the specificity of dementia (PLWD) in the introduction. It would be good to capture the specificity of the functioning of patients and the difficulties experienced by their families. It seems important to me to outline, at least in the theoretical layer, other possible predictors of the effectiveness of using the offered telephone help - those that could be attempted to be identified in future studies.

I believe that qualitative data, consisting of statements from people using the indicated help, would also contribute a lot. They could better illustrate the problem. The methodological part requires some additions. The selection of the sample and presentation of the results do not raise any objections. It seems to me that the PROMIS Self-Efficacy inManaging Emotions Short Form 8A research tool has been described quite poorly and in this form it is difficult to draw conclusions about its reliability. The authors have correctly identified the limitations of the research. However, I think that the discussion of the results is worth expanding (after previously expanding the theoretical part).

**Do you want your identity to be public for this peer review?** For information about this choice, including consent withdrawal, please see our Privacy Policy

Reviewer #1: No

Reviewer #2: No

---

## [Author Response · Author response to Decision Letter 1]

13 Apr 2025

Editor review:

1. In the ethics statement in the Methods, you have specified that verbal consent was obtained. Please provide additional details regarding how this consent was documented and witnessed, and state whether this was approved by the IRB.

Verbal consent was documented by Helpline agents in their Personify system. This protocol was approved by IRB. This has been amended as suggested in the Methods section.

No funding-related text appears in the manuscript.

3. Please state what role the funders took in the study. (Funding Statement)

As we now note the funders had no role in study design, data collection and analysis, decision to publish, or preparation of the manuscript. The funder provided support in the form of salaries for coauthors, but did not have any additional role in the study design, data collection and analysis, decision to publish, or preparation of the manuscript. The specific roles of these authors are articulated in the ‘author contributions’ section. We also provide the revised Competing Interests Statement in the Manuscript. We confirm that this commercial affiliation does not alter our adherence to all PLOS ONE policies on sharing data and materials.

4. We have uploaded the minimal anonymized data set necessary to replicate study findings.

Thank you. This has been amended.

Reviewer #1:

Exclusions are (or seem) intuitive, but it would be nice to know how many were excluded, and/or to “operationalized/define “crisis situation” and “frequent repeat caller” a bit more. If this is left to the discretion of the Helpline agent, then none of that information may be available, but a sentence/phrase with that explanation could be added to the manuscript.

Thank you for this suggestion. The number of participants excluded was not documented by the Alzheimer’s Association Helpline agents. Crisis callers and frequent repeat callers have now been defined.

The first paragraph of the discussion could probably be tightened up a bit: “A score of 40 is 1 standard deviation lower than US general population and score of 60 is 1 SD higher than the US general population.” is redundant with the immediately preceding sentence and could just be removed. The subsequent three sentences, “Callers who requested three calls […]” could be condensed something like this: “Compared to that normative data, out caregivers had below-average emotional self-efficacy, especially when requesting multiple consultations (one: 47.1, two: 44.5, three: 39.4).”

We appreciate this comment and have amended this section.

“Our first finding indicates that those who later requested additional calls had the lowest levels of self-efficacy in managing emotions at baseline, indicating the highest level of need.” My sense is that “highest level of need” extends beyond that PROMIS can measure. There is more to “need” than emotional self-efficacy, after all. Something like “[…] suggesting an elevated level of need” might better capture the scope of the present work.

Thank you. This has been updated with the suggested language.

This paragraph is more of a result than a discussion; the authors should consider moving to another section, or abbreviating or eliminating it, since the information it contains is a retelling of the contents of Table 3:“Five items from the PROMIS Self-Efficacy measure were significant in those who had one or two care consultations. The items “I can handle negative feelings”, “I can find ways to manage stress”, “I can avoid feeling discouraged stress”, and “I can keep emotional distress from interfering with things” were significantly improved (p <0.01) in those who had one care consultation. The item “I can find ways to manage stress” was significantly improved from baseline in those who had two care consultations” (p<0.01).”

Thank you. This has been moved to Results section and briefly summarized in the Discussion section.

This study has several limitations. First […]” The authors name only one limitation before proceeding to the next paragraph (“Despite these limitations […]”). …did the authors intend for the later paragraph (“Practice recommendations cannot […]”) to be a second limitation? Another one would be the loss-to-follow-up of almost half the sample (2503 with initial consultation, 1375 of those engaging with the call two weeks later). Another would be that those engaging with subsequent calls may be a self-selected group of people who find benefit from the consultation calls.

Thank you. The Limitations section has been updated with this information

“Our second finding suggests that callers benefited from all calls […]” are the authors sure they want to commit to *all*? Every single one? What if some of those 2503 didn’t value the call, and that’s why they were lost to follow-up?

Thank you. This has been amended.

Reviewer 2:

First of all, I propose expanding the theoretical background. Although there are few comparable studies, it would be worthwhile to outline in a more exhaustive way the difficult and problematic situations that result from the specificity of dementia (PLWD) in the introduction.

Thank you. We now provide further elaboration of the specific difficulties inherent in dementia caregiving.

It would be good to capture the specificity of the functioning of patients and the difficulties experienced by their families. It seems important to me to outline other possible predictors of the effectiveness of using the offered telephone help - those that could be attempted to be identified in future studies.

Thank for your this helpful comment.We have added a discussion of other possible predictors to our concluding statements on future research.

I believe that qualitative data, consisting of statements from people using the indicated help, would also contribute a lot. They could better illustrate the problem.

Unfortunately, we do not currently have any recorded/documented qualitative data. We have noted this in the manuscript as a limitation.

The methodological part requires some additions. The selection of the sample and presentation of the results do not raise any objections. It seems to me that the PROMIS Self-Efficacy in Managing Emotions Short Form 8A research tool has been described quite poorly and in this form it is difficult to draw conclusions about its reliability.

Thank you. More information has been added about the PROMIS instrument and its reliability.

There are possibly not a lot of studies about the non-inferiority of training methods like yours that extend the findings to outcomes for clients eg CG and PLWD, but rater stop at measuring knowledge and confidence etc. This is worth highlighting here somewhere also or in the discussion perhaps.?

Similar to our comment above, we now highlight how this study protocol will evaluate outcomes beyond typical training outcomes of “confidence” and “knowledge”.

The authors have correctly identified the limitations of the research. However, I think that the discussion of the results is worth expanding

We appreciate this comment, and we have expanded our discussion of results.

---

## [Decision Letter · Decision Letter 1]

19 May 2025

Dear Dr. Hodgson,

 Authors have addressed reviewer's concerns. #However, this work was published as an abstract in December 2023 https://alz-journals.onlinelibrary.wiley.com/doi/epdf/10.1002/alz.082552 that needs to be duly acknowledged and cited as feasible. Even though it was your own work and presented in part- missed that important piece may be misconstrued as omission. File is attached for your reference.  #Also, in the abstract 2504 participants were included that is 2503 in this manuscript- any reason for this discrepancy? #Abstract can include more information in methods section (some mention of statistical analyses or anything relevant in methodology)  and results (Additional relevant information). 

We look forward to receiving your revised manuscript.

Kind regards,

Vineet Gupta, MD, FACP, SFHM, CHCQM

Academic Editor

PLOS ONE

Journal Requirements:

Reviewers' comments:

Reviewer's Responses to Questions

**Comments to the Author**

Reviewer #2: All comments have been addressed

2. Is the manuscript technically sound, and do the data support the conclusions?

Reviewer #2: Yes

3. Has the statistical analysis been performed appropriately and rigorously?

Reviewer #2: Yes

4. Have the authors made all data underlying the findings in their manuscript fully available?

Reviewer #2: Yes

5. Is the manuscript presented in an intelligible fashion and written in standard English?

Reviewer #2: Yes

Reviewer #2: Dear Editors,

First of all, I would like to note that the author's article with the same title has been published in the meantime in the journal Alzheimer's & Dementia 19(S19),December 2023 DOI: 10.1002/alz.082552. I have not had the opportunity to compare the content (from the abstract I conclude that they may concern exactly the same study), but the fact that a text with the same title exists is the basis for changing the title.

The authors have addressed my earlier comments and in this respect I accept the changes.

**Do you want your identity to be public for this peer review?** For information about this choice, including consent withdrawal, please see our Privacy Policy

Reviewer #2: No

---

## [Author Response · Author response to Decision Letter 2]

14 Aug 2025

Please find our response to editor and reviewer comments below:

This work was published as an abstract in December 2023 https://alz-journals.onlinelibrary.wiley.com/doi/epdf/10.1002/alz.082552 that needs to be duly acknowledged and cited as feasible. Even though it was your own work and presented in part- missed that important piece may be misconstrued as omission. File is attached for your reference. 

Thank you for your attention to this. We presented early data from this study in 2023. We did not use the full dataset and these were preliminary findings presented as an abstract.

Also, in the abstract 2504 participants were included that is 2503 in this manuscript- any reason for this discrepancy?

We appreciate your attention to this detail. The work presented in the abstract was preliminary and did not include the final dataset. For the final analysis, we identified rules for duplicates and data missingness and this changed the number of included participants in this final analysis.

Abstract can include more information in methods section (some mention of statistical analyses or anything relevant in methodology) and results (Additional relevant information). 

Thank you for this feedback. We have updated the abstract with additional information in these sections.

Please review your reference list to ensure that it is complete and correct.

We have reviewed the reference list. We removed a citation because it was no longer cited in the text.

First of all, I would like to note that the author's article with the same title has been published in the meantime in the journal Alzheimer's & Dementia 19(S19),December 2023 DOI: 10.1002/alz.082552. I have not had the opportunity to compare the content (from the abstract I conclude that they may concern exactly the same study), but the fact that a text with the same title exists is the basis for changing the title. Change title (minor change)

Thank you for this recommendation. We have now changed the title to “Beyond Single Support Calls: Efficacy of Follow-Up Callbacks for Dementia Caregivers Using the Alzheimer's Association Helpline”.

---

## [Editor Report · Decision Letter 2]

16 Sep 2025

Beyond Single Support Calls: Efficacy of Follow-Up Callbacks for Dementia Caregivers Using the Alzheimer's Association Helpline

PONE-D-24-41264R2

Dear Dr. Hodgson,

We’re pleased to inform you that your manuscript has been judged scientifically suitable for publication and will be formally accepted for publication once it meets all outstanding technical requirements.

Kind regards,

Vineet Gupta, MD, FACP, SFHM

Academic Editor

PLOS ONE
---

## [Editor Report · Acceptance letter]

PONE-D-24-41264R2

PLOS ONE

Dear Dr. Hodgson,

I'm pleased to inform you that your manuscript has been deemed suitable for publication in PLOS ONE. Congratulations! Your manuscript is now being handed over to our production team.

Kind regards,

on behalf of

Dr Vineet Gupta

Academic Editor

PLOS ONE